# HASTE: Hardware-Aware Dynamic Sparse Training for Large Output Spaces

**Nasib Ullah** [1]  **Jinbin Zhang** [1]  **Jean Lucien Randrianantenaina** [2]  **Erik Schultheis** [3]  **Rohit Babbar** [2]

## Abstract

Extreme multi-label classification (XMC) involves learning models over large output spaces with millions of labels, making the output layer a memory-compute bottleneck. While sparsity-based methods reduce arithmetic complexity, they often fail to yield proportional speedups due to irregular memory access, poor hardware utilization, or reliance on auxiliary architectural components in long-tailed regimes. We introduce group-shared fixed fan-in sparsity, a semi-structured output-layer design in which semantically related labels share a sparse input pattern while retaining independent weights. This grouping introduces a task-aligned inductive bias—encouraging related labels to share feature subsets—while reducing index memory overhead, increasing feature reuse across labels, and enabling efficient GPU execution via custom CUDA kernels that leverage modern accelerator primitives. As an alternative to auxiliary objectives, we exploit the long-tailed structure of XMC by decomposing the output layer into a small dense head over frequent labels and a group-shared sparse tail over the remainder, providing an informative gradient pathway while preserving the memory benefits of sparsity. Through kernel-level microbenchmarking, we show that group-shared fixed fan-in translates arithmetic reductions into practical wall-clock gains, achieving up to $4.4\times$ speedup in the forward pass and up to $25\times$ speedup in backward passes over standard fixed fan-in sparsity, while operating within a few percent of a FLOPs-matched dense bottleneck. Across large-scale XMC benchmarks, our approach matches or improves precision@k over prior sparse baselines, while narrowing the performance gap to dense.

[1]Department of Computer Science, Aalto University, Espoo, Finland [2]Department of Computer Science, University of Bath, Bath, UK. [3]IST Austria. Correspondence to: Nasib Ullah <nasibullah.nasibullah@aalto.fi>.

*Proceedings of the $43^{rd}$ International Conference on Machine Learning*, Seoul, South Korea. PMLR 306, 2026. Copyright 2026 by the author(s).

## 1. Introduction

In many real-world applications, such as tagging, product-to-product recommendation, or matching search queries to advertisements, the output space can be extremely large. Extreme multi-label classification (XMC) (Bhatia et al., 2016; Babbar & Schölkopf, 2017; 2019), which trains classifiers over millions of labels, has therefore become central to several high-impact industrial systems. However, the enormous memory and computational demands, particularly that of the output layer of XMC models, has motivated a decade of research into more scalable training strategies. Early work explored label-tree-based dynamic negative sampling (Jiang et al., 2021; Zhang et al., 2021; Kharbanda et al., 2022) and nearest-neighbor-based multistage methods (Dahiya et al., 2021; 2023a;b) to reduce computational costs; however, these approaches largely left memory requirements unchanged.

More recently, memory-efficient techniques such as skip-loss optimization (Jain et al., 2023) and training with low-precision via quantization (Zhang et al., 2025) have been proposed to train state-of-the-art XMC models on consumer GPUs. As an alternative strand, there has been significant progress in training sparse neural networks (Lasby et al., 2024; Evci et al., 2020). However, modern GPU architectures are optimized for dense, contiguous memory blocks; consequently, unstructured sparsity results in low hardware utilization and divergent execution paths. Therefore, most recent approaches for training sparse neural networks emphasize the need to simulate sparsity using dense masks (Gadhikar et al., 2026). However, an off-the-shelf application of these methods, simulating sparsity via dense masks, cannot potentially achieve any realistic savings in memory-intensive XMC settings.

There have been attempts to find a middle ground between the two ends, (i) ill-suited nature of sparsity to GPU architecture and (ii) memory demands of dense (masked) training. In particular, a careful introduction of structure in the sparsity pattern via fixed fan-in per label in the output layer of XMC pipelines (Schultheis & Babbar, 2023) partially addresses the GPU load balancing. In this vein, an example of an end-to-end training approach is SPARTEX (Ullah et al., 2024), which adopts semi-structured fixed fan-in sparsity (Lasby et al., 2024), where each output neuron connects

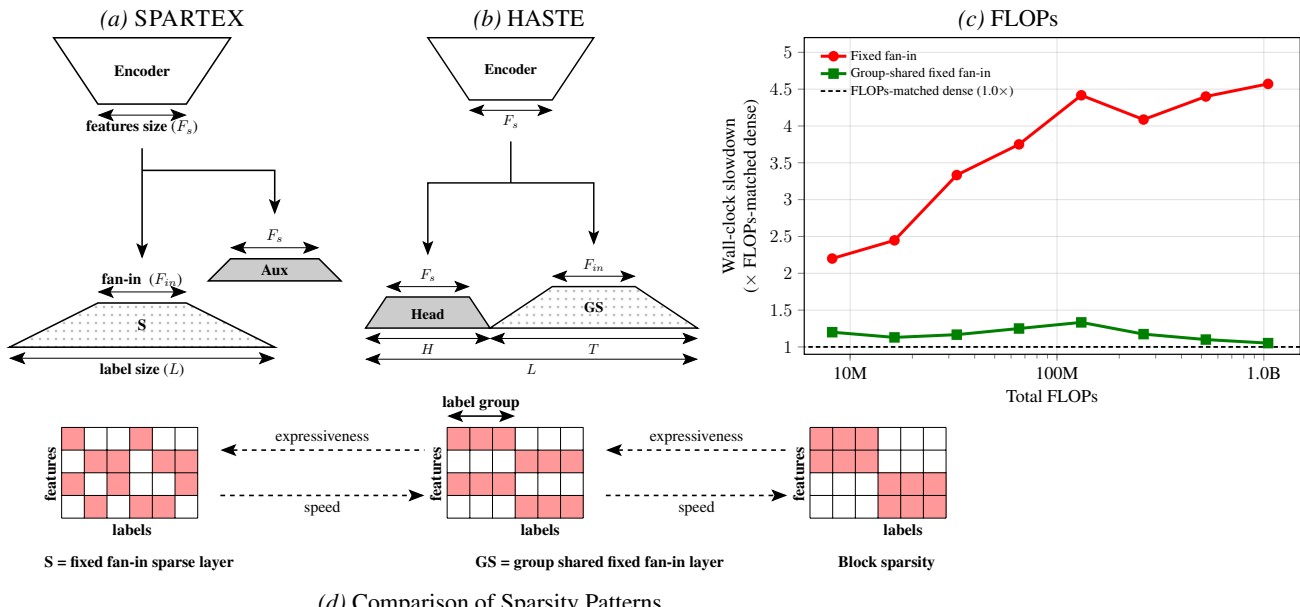

*Figure 1.* Architecture and Performance Overview. (a) Original SPARTEX. (b) Proposed Group-Shared architecture. (c) Performance benchmarks vs. FLOPs-matched dense (bottleneck) baseline. (d) Visual comparison of sparsity patterns showing trade-offs between expressiveness and speed.

to a fixed number of input features. This design reduces representational memory overhead and provides uniform load balancing across labels.

Approaches such as SPARTEX, though successful in reducing the memory footprint, still face two fundamental limitations. **Firstly**, fixed fan-in sparsity per-label remains fundamentally memory-bound. For each label, it issues a fixed number of random, uncoalesced memory reads; although computation is parallelized across labels, the lack of shared feature access quickly saturates memory bandwidth on modern accelerators—an instance of the well-known memory wall (Wulf & McKee, 1995). This access pattern further limits utilization of Tensor Core units central to modern ML hardware. Improving efficiency, therefore, requires introducing structure into sparse feature access. **Secondly**, training in long-tailed regimes is prone to instability, as sparse connectivity provides weak and noisy gradient signals to the encoder. SPARTEX introduces an auxiliary objective that provides an additional gradient pathway to the encoder (Figure 1.(a)). While effective in practice, this approach relies on auxiliary supervision that is not universally available and can be brittle: mismatches between auxiliary and primary objectives may introduce conflicting gradients, and auxiliary losses often require careful tuning. Moreover, in large-label regimes, the auxiliary branch can also incur nontrivial computational and memory overhead.

In order to tackle the **first limitation** of memory-boundedness due to irregular memory accesses, two natural

directions emerge: (i) increasing regularity to enable coalesced memory reads, and (ii) increasing feature reuse across labels to amortize data movement. Block sparsity (Okanovic et al., 2025) satisfies both by assigning contiguous blocks of features as the fan-in and sharing these blocks across groups of labels, yielding excellent hardware efficiency (Figure 1.(b)). However, the strong regularity constraints imposed by block sparsity requiring features to be both contiguous and shared, significantly restrict model expressiveness and can degrade performance.

In this work, we propose HASTE, a more flexible alternative based on group-shared fixed fan-in sparsity. Instead of assigning separate random fan-in locations to each label, we first form label groups, e.g., using semantic-based similarity, and assign each group a shared but randomly selected fan-in. This design preserves the benefits of feature reuse within label groups while enabling coalesced memory access patterns compatible with modern Tensor Core hardware. It also reduces representational memory overhead by a factor proportional to the group size. Importantly, it is also conceptually grounded in the structure of tasks: in applications such as product recommendation, semantically related products naturally benefit from accessing similar subsets of features. As shown in Figure 1.(b), our approach sits between semi-structured fixed fan-in sparsity and fully regular block sparsity, striking a practical balance between latency and expressiveness. Figure 1.(c) quantifies this trade-off via wall-clock slowdown measured relative to a FLOPs-matched dense bottleneck.

To address the **second limitation** of the optimization challenges of training in long-tailed label distributions with sparse signals, we adopt a simpler, data-driven alternative by decomposing the label space itself. Specifically, we partition the label set into two disjoint subsets and model a small subset of frequent labels using a dense classifier, while handling the remaining (and much larger) subset with a sparse output layer (Figure 1.(b)). This design leverages the inherently long-tailed nature of XMC datasets (Jain et al., 2016; Schultheis et al., 2023): by assigning only the most frequent labels to the dense component, we provide a consistent source of gradient signal to the encoder during training. At the same time, modeling the vast tail with a sparse layer preserves the memory and computational efficiency required at scale. To summarize, this paper makes the following contributions:

- We propose HASTE, based on group-shared fixed fan-in sparsity for large output layers, where semantically related labels share a common sparse feature pattern. This design aligns with task structure, increases arithmetic intensity, and enables efficient use of modern GPU primitives such as Tensor Cores.

- We introduce an auxiliary-free optimization strategy by decomposing the label space into a small dense head over frequent labels and a sparse tail over the remaining labels, providing a consistent gradient signal while preserving the memory benefits of sparsity.

- Across large-scale XMC benchmarks, our approach outperforms prior sparsity-based baselines and substantially narrows the gap to dense models, remaining competitive or superior at multi-million–label scales.

- Kernel-level microbenchmarks show that group-shared fixed fan-in forward and backward kernels achieve significant speedups and operate close to FLOPs-matched dense baselines, translating sparsity-induced FLOPs reductions into real wall-clock gains.

## 2. Related Work

### 2.1. Extreme Classification

Earlier work in extreme classification (Khandagale et al., 2020; Prabhu et al., 2018) reduced computation through label partitioning and tree-based models, which limit prediction and training to a small subset of labels via hierarchical organization or negative sampling. Subsequent methods (You et al., 2019; Chang et al., 2020; Jiang et al., 2021; Zhang et al., 2021; Kharbanda et al., 2022) incorporated deep encoders into these pipelines, or employed nearest-neighbor retrieval and multi-stage training (Dahiya et al., 2021; 2023a;b) to decouple representation learning

from classification. These methods typically trade full-label scoring for candidate generation and reranking, making their efficiency depend on retrieval quality.

While effective in reducing arithmetic cost, these approaches do not fundamentally address the memory footprint of the output layer. More recent end-to-end methods such as RE-NEE (Jain et al., 2023), ELMO (Zhang et al., 2025), and SPARTEX (Ullah et al., 2024) address output-layer scalability more directly through memory-efficient losses, low-precision training, or sparse classifiers. This distinction is important in million-label regimes, where data movement and output-layer storage can dominate practical training efficiency. In contrast to previous works, by revisiting sparsity from a hardware-aware perspective, we focus on output-layer designs that simultaneously improve memory efficiency and practical training throughput at extreme label scales.

### 2.2. Sparse Training and Hardware-Efficient Sparsity

Unstructured sparsity (Mocanu et al., 2018; Evci et al., 2020) is inefficient on modern GPUs due to irregular memory access and poor utilization of vectorized compute units (Gale et al., 2019; Hooker, 2021); consequently, many sparse training methods simulate sparsity using dense masks, limiting its impact on training efficiency. To address this, structured sparsity patterns such as N:M sparsity (Zhou et al., 2021; Lin et al., 2023; Castro et al., 2023) leverage specialized hardware support, including sparse Tensor Cores. While highly efficient, these patterns such as V:N:M (Castro et al., 2023) and BLOCK-SPARSE (Okanovic et al., 2025) either impose rigid constraints or typically limited to moderate sparsity ratios (Zhou et al., 2021; Hu et al., 2024), restricting their applicability.

Fixed fan-in sparsity (Lasby et al., 2024; Schultheis & Babbar, 2023; Ullah et al., 2024) provides a flexible alternative by enforcing a constant number of incoming connections per output while allowing arbitrary feature selection. This ensures uniform computational load and low representation overhead, making it attractive for large output spaces. However, because fan-in features are selected independently per output, memory accesses remain random and uncoalesced, leaving fixed fan-in sparsity fundamentally memory-bound and limiting effective utilization of modern accelerator primitives such as Tensor Cores.

In this work, we propose group-shared fixed fan-in sparsity, which introduces structured feature reuse by allowing semantically related outputs to share a common sparse input pattern while retaining independent weights. This design reduces index memory overhead, improves memory locality, and enables efficient GPU kernels that leverage modern accelerator primitives, while remaining substantially more expressive than other alternatives.

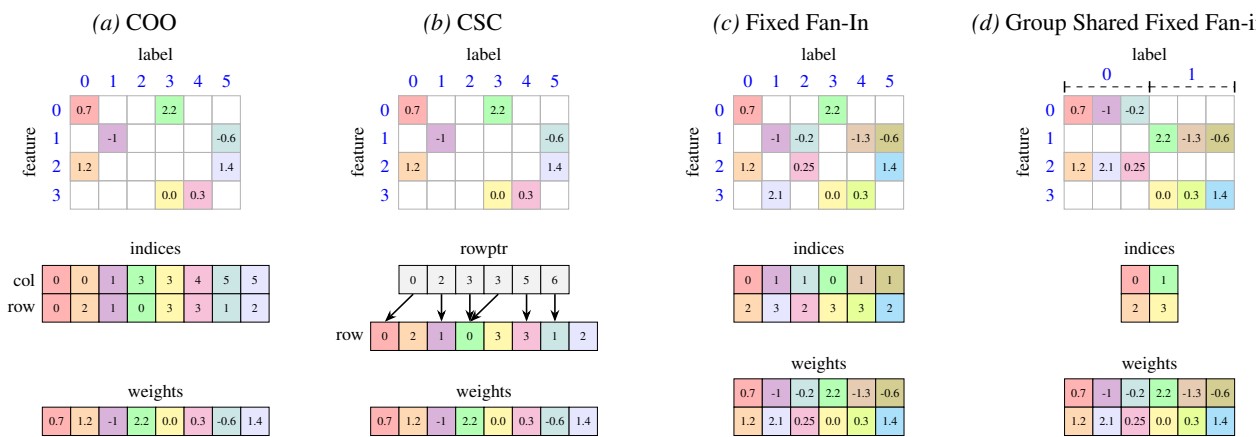

*Figure 2.* Comparison of Group-shared Fixed fan-in format with Fixed Fan in sparsity and standard Compressed sparse column (CSC) and COO format

## 3. Hardware-Aware Sparse Training

**Problem Setup.** We consider extreme multi-label classification with $L$ labels and an encoder $f_\theta$ that maps an input $x$ to a feature representation $h = f_\theta(x) \in \mathbb{R}^H$. Predictions are produced by a linear output layer with weights $W \in \mathbb{R}^{L \times H}$, where $L$ can be on the order of millions. Our goal is to design a sparse output-layer structure that reduces memory and computation while preserving predictive performance and enabling efficient GPU execution.

### 3.1. Group Shared Fixed Fan-in Sparsity

**Representation.** The fixed fan-in sparsity format (Figure 2.(c)) enforces a constant number of incoming connections per label, which yields uniform computational load across outputs. Compared to standard sparse matrix formats such as CSR or CSC (Figure 2.(b)), fixed fan-in requires lower representation overhead, as it stores a single index array without row pointers. Group-shared fixed fan-in sparsity (Figure 2.(d)) builds on this structure by allowing a group of labels to share the same fan-in pattern.

In group-shared fixed fan-in sparsity, the labels are partitioned into groups $\{\mathcal{G}_k\}_{k=1}^K$, such that $|\mathcal{G}_k| =: G$. Each group $\mathcal{G}_k$ chooses a support set $\mathcal{I}_k \subseteq [H]$ with $|\mathcal{I}_k| =: F$ ("fixed fan-in per label group"). Each label $\ell \in \mathcal{G}_k$ has parameters $w_\ell \in \mathbb{R}^F$ living only on $\mathcal{I}_k$. The logit corresponding to a label $\ell$ for an instance $x$ is obtained as $z_\ell(x) = \langle w_\ell, h_{\mathcal{I}_{g(\ell)}} \rangle$, where $w_\ell \in \mathbb{R}^F$ is the per-label weight vector and $g(\ell)$ maps label $\ell$ to its group.

**Index overhead for sparse representations.** For $L$, $F$, and $G$ as defined earlier, index storage for COO requires $2LF$ indices, while CSR/CSC requires $LF + (L + 1)$ indices. Fixed fan-in sparsity in SPARTEX (Ullah et al., 2024) stores $LF$ indices, eliminating row pointers entirely. In contrast, group-shared fixed fan-in sparsity stores only $(L/G)F$ in-

dices, reducing index memory by approximately a factor of $G$ relative to standard fixed fan-in sparsity. As a result, this choice drastically reduces the indexing overhead as the fan-in indices are stored once per group rather than once per label, and thereby saving precious GPU memory. Figure 3 illustrates the index memory overhead as a function of the number of labels $L$ for different sparsity formats, highlighting the substantially lower growth rate of group-shared fixed fan-in sparsity in extreme output spaces.

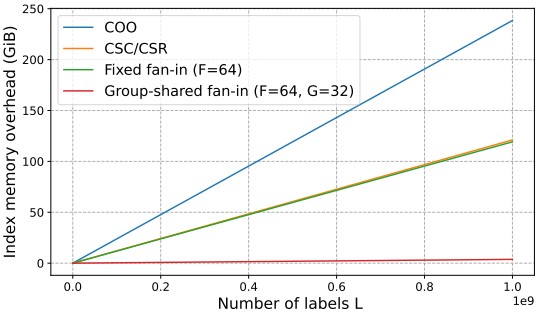

*Figure 3.* Indexing memory overhead as label size increases for different forms of sparsity.

**Label Grouping.** Labels in a group are merged together to ensure that semantically related labels belong to the same group. Since group-shared fixed fan-in sparsity requires labels within a group to share a common fan-in index set, these group-wise labels are encouraged to attend to similar subsets of input features. Concretely, this is achieved by constructing groups $\{\mathcal{G}_k\}_{k=1}^K$ as follows :

$$\{\mathcal{G}_k\}_{k=1}^K = \arg \max_{\text{partition}} \sum_{k=1}^K \sum_{\ell \in \mathcal{G}_k} \text{sim}(e_\ell, \mu(\mathcal{G}_k)), \quad (1)$$

where $e_\ell \in \mathbb{R}^H$ is a label embedding, $\mu(\mathcal{G}_k)$ is the group

centroid, and $\text{sim}(\cdot, \cdot)$ denotes cosine similarity. We compute $e_\ell$ in a data-driven manner without training a dense classifier. Let $\mathcal{P}_\ell = \{i : y_{i\ell} = 1\}$ be the set of training instances where label $\ell$ is positive and let $h_i = f_\theta(x_i) \in \mathbb{R}^H$ be the encoder representation. These embeddings, which are reusable across runs, can be computed offline as :

$$e_\ell = \text{Normalize}\left(\frac{1}{|\mathcal{P}_\ell|} \sum_{i \in \mathcal{P}_\ell} h_i\right). \qquad (2)$$

Solving (1) exactly is intractable when $L \sim \mathcal{O}(10^6)$, so we adopt a scalable two-stage approximation. First, we apply mini-batch spherical $k$-means to partition labels into $C$ coarse clusters, where $C \approx L/(\beta G)$ and $\beta$ controls the average bucket size. Second, within each coarse cluster, we form groups greedily: we sample an unassigned seed label and select its top-$(G-1)$ nearest unassigned neighbors (by cosine similarity) to form $\mathcal{G}_k$. This yields compact, non-overlapping groups of approximately uniform size. The detailed algorithm is provided in Algorithm 2. Label grouping defines a fixed permutation of labels such that labels within the same group are contiguous. In practice, this only requires remapping label identifiers to output neuron indices.

### 3.2. Kernels for Group-shared Fixed Fan-in Sparsity

Tying the feature location over label-groups induces a regular, group-aligned computation pattern, which we exploit in custom CUDA kernels. Concretely, with label groups $\mathcal{G}_k$ such that $|\mathcal{G}_k| = G$ and shared fan-in index set $\mathcal{I}_k$ for all labels in $\mathcal{G}_k$ and labels within a group are laid out contiguously, this structure enables a "gather-once and dense MMA" execution pattern. Each thread block is tiled over the batch and label dimensions, with the tile size along the label dimension chosen as a multiple of the group size $G$. In the forward pass, for a batch tile $B_t$ and label group $\mathcal{G}_k$, let $H_k = h_{:,\mathcal{I}_k} \in \mathbb{R}^{B_t \times F}$ be the gathered-at-once features from the input batch $B_t$ and $W_k \in \mathbb{R}^{G \times F}$ which represents the weights to be learnt for the label group over the corresponding fan-in index set $\mathcal{I}_k$, the core compute inside a block is a dense GEMM on the output tiles $Z_k = H_k \times W_k^T \in \mathbb{R}^{B_t \times G}$, which is exactly the computation Tensor Core MMA targets. Notably, when compared to SPARTEX which has fixed fan-in on a *per label basis*, there is no shared input tile $H_k$ and core computation is thin vector dot-products (instead of Tensor-core friendly) with lots of data movement with low reuse. On the other hand, our approach allows the shared fan-in index set $\mathcal{I}_k$ and $H_k$ for a group to be staged once in fast shared memory and reused by all warps in the block.

The gradient with respect to the group weights $W_k$ is computed analogously to the forward pass. For each group $\mathcal{G}_k$, the backward-weight computation corresponds to a dense matrix multiplication restricted to the fan-in features be-

tween the output gradients and the gathered feature tile, i.e., $\nabla W_k = (\nabla Z_k)^\top H_k$, where $H_k = h_{:,\mathcal{I}_k} \in \mathbb{R}^{B_t \times F}$ contains only the fan-in features for group $\mathcal{G}_k$. Gradients are computed exclusively on this fan-in support, and no dense $H$-dimensional gradients are formed.

Computing gradients with respect to the input features is more challenging, as each feature dimension accumulates contributions from multiple labels and groups. Specifically, the backward-feature computation involves a reduction over the label dimension, where partial gradients of the form $\nabla h_{:,\mathcal{I}_k} \mathrel{+}= \nabla Z_k W_k$ are aggregated across all groups whose fan-in includes the same feature indices. To parallelize this reduction, we employ a Split-$K$ strategy (Hoque et al., 2024) over the label dimension. Each thread block computes partial feature gradients for a subset of label groups, which are accumulated to form the final feature gradients. Since extreme classification operates in regimes where the label dimension scales to millions, backward-feature computation is dominated by large label-wise reductions, which are especially costly under irregular or poorly aligned sparsity patterns; in our approach, the label group size $G$ governs parallelism along the label dimension, improving load balance and mitigating serialization in large-label regimes, as reflected in the group-size versus wall-clock time results (Figure 5).

### 3.3. Head–Tail Split for Training Stability

To stabilize training in extreme long-tailed settings, we adopt a head–tail split. Let $\mathcal{Y} = \mathcal{H} \cup \mathcal{T}$, with $\mathcal{H} \cap \mathcal{T} = \emptyset$, denote the split of labels based on frequency. As shown in Figure 1.(b), frequent labels in $\mathcal{H}$ are handled by a small dense, while infrequent labels in $\mathcal{T}$ use the proposed group-shared fixed fan-in sparsity. Both branches share the same encoder $h = f_\theta(x)$ but apply separate lightweight feature projections, $h_{\text{head}} = P_{\text{head}} h$ and $h_{\text{tail}} = P_{\text{tail}} h$, before their respective output layers. This design is fully compatible with group-shared fixed fan-in sparsity and does not affect the grouping or kernel implementations.

For $n$ training samples, we optimize the encoder parameters $\theta$, the dense-head weights $W_{\text{head}}$, the tail weights $\{w_\ell\}_{\ell \in \mathcal{T}}$, and the group fan-in assignments $\{\mathcal{I}_k\}_{k=1}^K$. Let $\Theta$ denote this collection. We define $a_{i\ell}^{\mathcal{H}} = (W_{\text{head}} h_{\text{head}}(x_i))_\ell$ and $a_{i\ell}^{\mathcal{T}} = \langle w_\ell, h_{\text{tail}}(x_i)_{\mathcal{I}_{g(\ell)}} \rangle$. The end-to-end objective is

$$\min_\Theta \frac{1}{n} \sum_{i=1}^n \left[ \sum_{\ell \in \mathcal{H}} \text{BCE}\big(y_{i\ell}, \sigma(a_{i\ell}^{\mathcal{H}})\big) \right.$$
$$\left. + \sum_{\ell \in \mathcal{T}} \text{BCE}\big(y_{i\ell}, \sigma(a_{i\ell}^{\mathcal{T}})\big) \right]. \qquad (3)$$

The training loop follows an alternating minimization procedure with (i) continuous phase (parameter fitting) in which the support $\{\mathcal{I}_k\}$ is held fixed, and we optimize $\theta$

*Table 1.* Comparison of the precision@k performance of our proposed method with state-of-the-art XMC methods on the AmazonTitles-670K, Amazon-670K, Amazon-3M datasets, and, LF-Paper2Keywords dataset. **Bold** indicates the best results on sparse baselines, and underline indicates the best results among all baselines. $M_{tr}$ denotes peak training memory.

| Method | Sparsity | P@1 | P@3 | P@5 | $M_{tr}$ (GiB) | E. Time (mm:ss) | Sparsity | P@1 | P@3 | P@5 | $M_{tr}$ (GiB) | E. Time (mm:ss) |
|---|---|---|---|---|---|---|---|---|---|---|---|---|
| | | | AmazonTitles-670K | | | | | | Amazon-670K | | | |
| LIGHTXML | - | 41.6 | 37.2 | 33.9 | 15.0 | 19:02 | - | 47.3 | 42.2 | 38.5 | 12.4 | 53:30 |
| CASCADEXML | - | 42.1 | 37.5 | 34.1 | 22.3 | 11:32 | - | 48.5 | 43.7 | 40.0 | 18.3 | 16:46 |
| DENSE | - | 43.7 | 39.1 | 35.9 | 12.5 | 2:04 | - | 50.6 | 45.2 | 41.1 | 11.9 | 7:14 |
| DENSE BN | - | 41.3 | 36.8 | 33.7 | 3.9 | **1:42** | - | 43.8 | 39.2 | 35.9 | 4.0 | **5:32** |
| BLOCK SPARSE | 83 | 39.4 | 34.6 | 30.9 | 5.3 | 1:44 | 92 | 45.0 | 39.8 | 35.7 | 3.2 | 5:35 |
| SPARTEX | 83 | 42.6 | 38.2 | 35.1 | 5.0 | 4:44 | 83 | 47.1 | 41.8 | 38.0 | 3.7 | 8:01 |
| HASTE | 83 | **43.0** | **38.7** | **35.5** | **3.2** | 1:46 | 92 | **48.1** | **43.1** | **39.2** | **2.1** | 5:39 |
| | | | Amazon-3M | | | | | | LF-Paper2Keywords-8.6M | | | |
| LIGHTXML | - | - | - | - | OOM | - | - | - | - | - | OOM | - |
| CASCADEXML | - | 51.3 | 49.0 | 46.9 | 87.0 | 90:00 | - | - | - | - | OOM | - |
| DENSE | - | 52.6 | 49.7 | 47.4 | 39.7 | 29:58 | - | 43.6 | 32.13 | 26.06 | 105.64 | 61:23 |
| DENSE BN | - | 47.0 | 44.6 | 42.7 | 13.1 | **21:17** | - | 34.3 | 26.4 | 21.9 | 13.24 | **31:25** |
| BLOCK SPARSE | 92 | 27.9 | 25.1 | 23.5 | 10.1 | 23:56 | 92 | 22.8 | 17.2 | 13.7 | 17.6 | 34:32 |
| SPARTEX | 83 | 50.2 | 47.1 | 44.8 | 13.5 | 86:38 | 92 | 40.7 | 28.7 | 22.3 | 18.36 | 141:35 |
| HASTE | 92 | **52.5** | **49.5** | **47.4** | **5.67** | 21:39 | 92 | **47.5** | **36.2** | **29.8** | **12.5** | 35:03 |

and weights ($W_{head}$ and $\{w_\ell\}_{\ell \in \mathcal{T}}$), and (ii) discrete phase (rewiring) wherein for each group $k$, modify $\mathcal{I}_k$ while keeping $|\mathcal{I}_k| = F, \forall k$. The overall training procedure is summarized in Algorithm 1, and the group-shared rewiring mechanism is detailed in Algorithm 3.

# 4. Experimental Results

**Datasets.** We evaluate on large-scale XMC benchmarks with label sizes starting from 670K. Specifically, we consider Amazon-670K, AmazonTitles-670K, Amazon-3M, and LF-Paper2Keywords-8.6M. All datasets exhibit highly long-tailed label distributions. LF-Paper2Keywords-8.6M is obtained from a separate public source (Zhang et al., 2025), while the remaining datasets are drawn from standard XMC repositories (Bhatia et al., 2016). Further details are provided in Table 7.

**Baselines and Evaluation Metrics.** We compare against three classes of baselines: (i) sampling-based XMC methods (LIGHTXML (Jiang et al., 2021), CASCADEXML (Kharbanda et al., 2022)); (ii) end-to-end dense (RENEE (Jain et al., 2023)) and DENSE BN (bottleneck) baseline; and (iii) sparsity-based methods (Okanovic et al., 2025), including SPARTEX (Ullah et al., 2024). For fair comparison, all methods use the same encoder architecture. We report Precision@k and propensity-scored Precision@k (PSP@k), and measure kernel-level wall-clock time using Triton's benchmarking utilities. Additional baseline details and evaluation protocols are provided in the Appendix A

**Implementation Details.** The proposed group-shared fixed fan-in kernels are implemented in CUDA and integrated into PyTorch, while the remaining components are implemented using the PyTorch framework. Following common practice (Ullah et al., 2024; Jain et al., 2023; Zhang et al., 2025), we use separate optimizers for the encoder and the classification layer: Adam for the encoder and SGD with momentum for the final layer. Training uses binary cross-entropy loss and is performed end-to-end in BF16 precision. We employ a head–tail split with the top 2–5% labels modeled densely and apply dynamic sparse training with a fixed sparsity budget and periodic rewiring. Label groups are formed with group sizes of 16, 32, or 64. All experiments are conducted on a single NVIDIA A100 GPU. We focus on single-GPU efficiency; multi-GPU execution and communication overheads are orthogonal to the contributions of this work. Code is available at `https://github.com/xmc-aalto/haste`.

**Empirical Performance.** Table 1 shows the precision@k mainly p@1, p@3 and p@5 performance for datasets ranging from 670K to 8.6 million labels. $M_{tr}$ represents peak training memory and E. Time represents epoch time. Our proposed approach outperforms SPARTEX in prediction performance, memory and epoch time, and that too at higher sparsity ratio. Even though the label-wise fixed fan-in employed in SPARTEX is more expressive than group-shared fan-in, the better performance is potentially due to a combination of the dense head for a small number of head-labels (see Table 6). Also, among sparse baselines along with

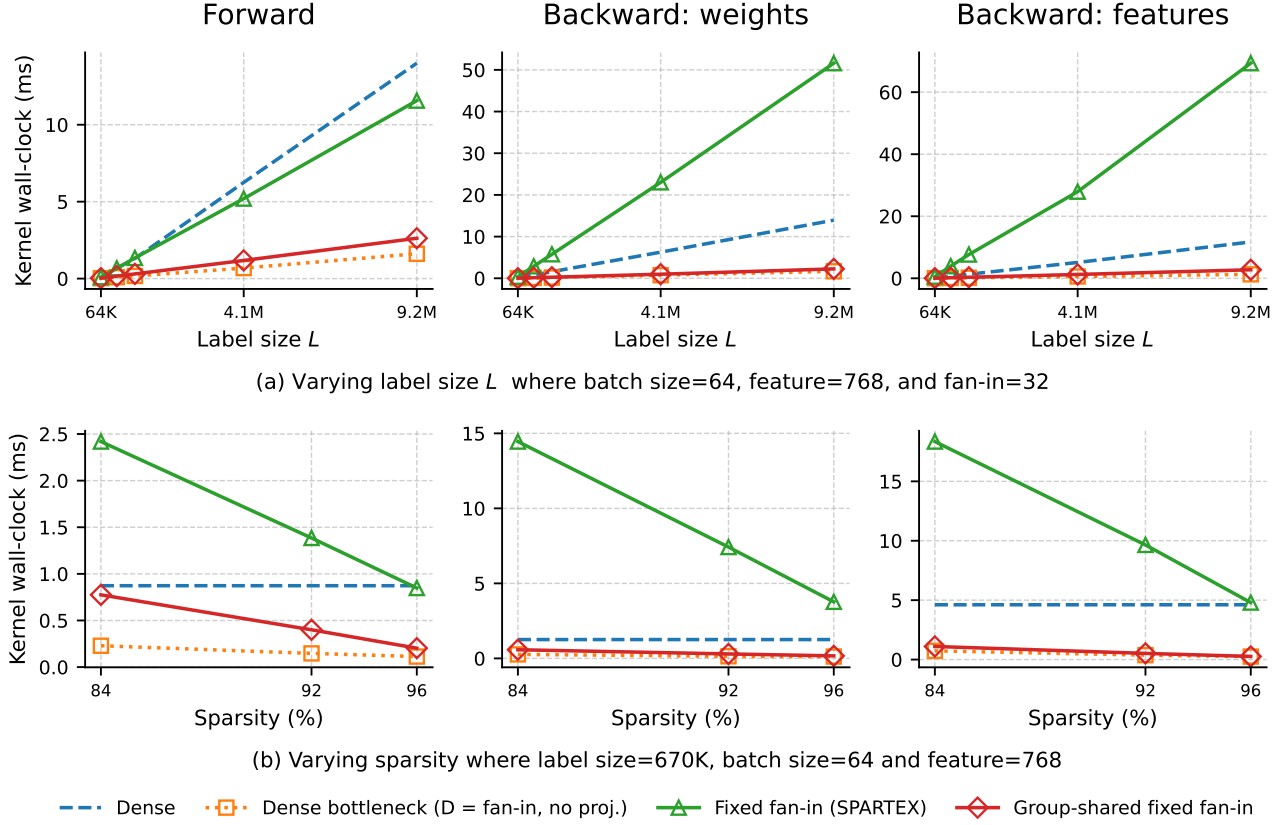

*Figure 4.* Kernel benchmarking of forward, gradient of weights and gradient of features for batch size 64, feature size 768, and, label group size of 32 on A100 GPU. (a) kernel wall clock time at different label size from 64K up to 9 millions for fan-in value of 32 (96% sparsity). (b) kernel wall clock time at different sparsity level from 83% to 96% at label size 670K.

dense bottleneck our method outperforms. Compared to dense performance we see our proposed approach decreases the gap compared to SPARTEX and the benefits are more as the label size increases. Additional results on label-feature datasets are provided in Appendix E.

**Kernel Microbenchmarking.** We microbenchmark the output-layer kernels to isolate compute effects. We compare group-shared fixed fan-in against standard fixed fan-in and dense matmul at fixed batch size and feature dimension, and include a FLOPs-matched dense baseline that operates directly on a reduced feature dimension equal to the fan-in value $F$. Figure 4.(a) plots kernel wall-clock time versus label size, while Figure 4.(b) varies sparsity at fixed label size. Notably, at moderate sparsity (e.g., 83%), standard fixed fan-in can be slower than dense matmul due to irregular, uncoalesced memory access, whereas group-shared fixed fan-in consistently outperforms fixed fan-in and remains close to the FLOPs-matched dense baseline—showing that structured feature reuse converts sparsity into practical wall-clock gains on GPUs.

**Label Grouping Strategies for Fan-in Sharing.** Group-

*Table 2.* Effect of label grouping strategies on group-shared fixed fan-in. Comparison of random, frequency-based, and semantic label grouping using Precision@k on Amazon-670K dataset.

| Label Grouping Strategy | P@1 | P@3 | P@5 |
|---|---|---|---|
| Random grouping | 46.3 | 41.7 | 37.9 |
| Frequency-based grouping | 46.7 | 42.0 | 38.1 |
| Semantic grouping | **48.1** | **43.1** | **39.2** |

shared fixed fan-in sparsity assumes that semantically related labels can effectively share a common fan-in pattern. While the overall precision@k results in Table 1 already suggest the benefits of this design, we explicitly evaluate the impact of different label grouping strategies. Table 2 compares three approaches on Amazon-670K: random grouping, frequency-based grouping, and semantic grouping based on label similarity. Semantic grouping consistently yields higher P@k, confirming that aligning fan-in sharing with label semantics improves representational quality.

**Performance on Tail Labels.** We evaluate tail-label performance using propensity-scored precision (PSP@k), which

*Table 3.* Comparison of the precision@k performance on the AmazonTitles-670K, Amazon-3M, and LF-Paper2Keywords-8.6M datasets. **Bold** indicates the best results on sparse baselines, and underline indicates the best results among all baselines.

| Method | PSP@1 | PSP@3 | PSP@5 | PSP@1 | PSP@3 | PSP@5 | PSP@1 | PSP@3 | PSP@5 |
|---|---|---|---|---|---|---|---|---|---|
| | AmazonTitles-670K | | | Amazon-3M | | | LF-Paper2Keywords-8.6M | | |
| RENEE | 27.0 | 31.1 | 34.89 | 14.39 | 17.47 | 19.80 | 3.3 | 3.5 | 4.0 |
| DENSE BN | 23.8 | 28.1 | 32.2 | 13.5 | 16.4 | 18.6 | 6.0 | 5.5 | 5.6 |
| SPARTEX | 24.1 | 28.0 | 31.6 | 14.3 | 17.2 | 19.4 | 4.0 | 3.6 | 3.5 |
| HASTE | **25.3** | **29.2** | **33.0** | **15.9** | **19.2** | **21.6** | **6.7** | **6.4** | **6.5** |

emphasizes rare labels. As shown in Table 3, our method consistently improves PSP@k over sparse baselines across datasets. Crucially, these gains indicate that the head–tail split improves training beyond merely increasing head-label capacity: despite allocating a dense head to frequent labels, PSP@k on the tail also improves, reflecting better gradient propagation and optimization in long-tailed regimes. On LF-Paper2Keywords-8.6M, our approach achieves the strongest PSP@k.

*Table 4.* Effect of label group size in group-shared fixed fan-in sparsity on Amazon-670K. Smaller group sizes consistently achieve higher Precision@k, highlighting the trade-off between representational capacity and hardware-friendly grouping.

| Group Size | P@1 | P@3 | P@5 |
|---|---|---|---|
| 16 | **48.1** | **43.1** | **39.2** |
| 32 | 47.7 | 42.6 | 38.7 |
| 64 | 47.5 | 42.3 | 38.3 |

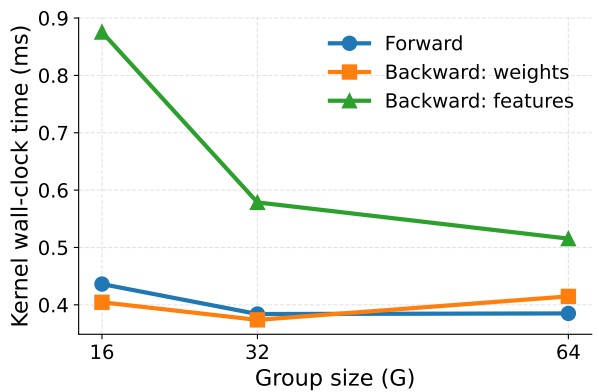

*Figure 5.* Label Group size vs Kernel wall clock time on A100 GPU.

**Effect of Label Group Size ($G$).** The group size $G$ controls the trade-off between representational flexibility and kernel efficiency. As shown in Table 4, smaller groups achieve slightly better accuracy: on Amazon-670K, $G$=16 yields

the highest P@k, while increasing $G$ to 32 or 64 leads to a modest but consistent drop. In contrast, larger groups improve kernel efficiency. Figure 5 shows that increasing $G$ substantially reduces the backward-feature kernel time, as more labels reuse the same gathered feature tile, amortizing memory access and exposing greater parallelism ( Split-$K$ over labels).

*Table 5.* Comparison with Quantization based methods

| Method | P@1 | P@3 | P@5 | $M_{\text{tr}}$ (GiB) | E. Time (mm:ss) |
|---|---|---|---|---|---|
| | Amazon-3M | | | | |
| ELMO(BF16) | **53.4** | **50.9** | **48.8** | 10.4 | 16:13 |
| ELMO(FP8) | 52.7 | 50.4 | 48.3 | 6.6 | 18:02 |
| HASTE | 52.5 | 49.5 | 47.4 | **5.67** | **14:11** |
| | LF-Paper2Keywords-8.6M | | | | |
| ELMO(BF16) | 45.4 | 33.6 | 27.2 | 18.8 | 33.16 |
| ELMO(FP8) | 43.4 | 31.6 | 25.4 | **9.0** | 36:42 |
| HASTE | **47.5** | **36.2** | **29.8** | 12.5 | **18:35** |

**Comparison with Quantization-Based Training.** Table 5 compares our method with quantization-based training using ELMO, with all sparse results operating at 92% sparsity. On Amazon-3M, ELMO with BF16 achieves the highest P@k; however, our approach also operates in BF16 while enforcing extreme sparsity, and remains competitive. On the larger LF-Paper2Keywords-8.6M dataset, our method exceeds ELMO, indicating favorable scaling at extreme label sizes. Beyond accuracy, pushing sparsity to very high regimes directly reduces data movement rather than numerical precision, which can be advantageous for memory- and energy-constrained settings. Finally, sparsity and quantization are complementary, and the proposed group-shared fixed fan-in structure is naturally compatible with low-precision arithmetic.

**Impact of Head–Tail Decomposition.** As shown in Table 6, introducing a dense head over frequent labels improves Precision@k for both our method and SPARTEX on Amazon-

670K, with our approach gaining $+1.3$ P@1. Crucially, these gains are not limited to head labels: the propensity-scored results in Table 3 show consistent improvements in PSP@k, indicating better gradient propagation and optimization for tail labels rather than a mere reduction in effective sparsity.

*Table 6.* Impact of head–tail (HT) decomposition on Amazon-670K.

| Method | P@1 | P@3 | P@5 |
|---|---|---|---|
| HASTE (NO HT) | 46.8 | 41.7 | 38.1 |
| HASTE (WITH HT) | **48.1** | **43.1** | **39.2** |
| SPARTEX | 47.1 | 41.8 | 38.0 |
| SPARTEX (WITH HT) | **47.6** | **42.2** | **38.2** |

## 5. Conclusion

We presented HASTE, a hardware-aware dynamic sparse training framework for XMC based on group-shared fixed fan-in sparsity that converts reductions in arithmetic complexity into proportional GPU speedups while preserving accuracy in large output spaces. By introducing structured feature sharing across semantically related labels and an optional head–tail decomposition, our approach improves both wall-clock efficiency and tail-label performance, substantially outperforming prior sparsity-based methods. Combining structured sparsity with low-precision training is a promising direction, and our Tensor Core–friendly design provides a natural foundation for jointly exploiting both forms of efficiency in future work.

## Impact Statement

This paper improves the efficiency of dynamic sparse training for models with very large output spaces by introducing a group-shared sparse output formulation and hardware-aware kernels that better match modern accelerators (e.g., Tensor Core–friendly tiled computation). In contexts such as extreme multi-label classification, these advances can reduce training time and memory footprint, potentially lowering energy use per experiment and making large-output models more accessible under fixed compute budgets.

However, greater efficiency can also lower the cost of deploying large-scale classifiers in sensitive applications. Moreover, extreme classification datasets are often long-tailed and imperfectly labeled, so efficiency gains do not address—and may amplify—existing biases or uneven performance on rare labels. Users should therefore evaluate tail-label and subgroup behavior explicitly and treat this method as complementary to, not a replacement for, responsible data curation, monitoring, and appropriate-use safeguards.

## Acknowledgements

We acknowledge the support of the Academy of Finland (Research Council of Finland) via grants 347707 and 348215 and the support of computational resources provided by the Aalto Science-IT project, and CSC IT Center for Science, Finland.

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

**Algorithm 1** Training with Group-Shared Fixed Fan-in Sparsity

---

**Require:** Training data $\mathcal{D}$, encoder $f_\theta$, labels $[L]$
**Require:** group size $G$, fan-in $F$, rewiring interval $\Delta T$
**Ensure:** Trained parameters
  Split labels into dense head and sparse tail
  Group tail labels into groups of size $G$ using Algorithm 2
  Initialize a shared fan-in support of size $F$ per group
  **for** $t = 1$ to $T$ **do**
    Sample a minibatch from $\mathcal{D}$ and compute features using $f_\theta$
    Compute predictions with the group-shared sparse classifier
    Compute loss and update model parameters
    **if** $t \bmod \Delta T = 0$ **then**
      Rewire group supports and weights using Algorithm 3
    **end if**
  **end for**

---

## A. Baselines and Evaluation Metrics

We compare our approach against representative deep XMC baselines built on transformer encoders, covering retrieval-based, tree-based, multi-stage, dense end-to-end, low-precision, and sparse-training paradigms:

- **LightXML** (Jiang et al., 2021): A transformer-based method that jointly trains a retriever and ranker, leveraging dynamic negative sampling to reduce computational cost during training.

- **CascadeXML** (Kharbanda et al., 2022): A hierarchical approach that decomposes extreme classification using a Probabilistic Label Tree (PLT), with different levels of the tree aligned to different layers of the transformer encoder.

- **SIAMESEXML** (Dahiya et al., 2021): A Siamese-network-based XMC method that learns representations for instances and labels and uses nearest-neighbor retrieval for scalable prediction.

- **NGAME** (Dahiya et al., 2023a): A multi-stage retrieval-based XMC method that improves scalable label prediction through representation learning and nearest-neighbor search.

- **DEXML** (Dahiya et al., 2023b): A deep extreme classification method that uses representation learning and retrieval/ranking stages to handle large label spaces.

- **Renee** (Jain et al., 2023): An end-to-end dense XMC model that introduces a loss shortcut to reduce memory overhead and employs a hybrid data-model parallel training strategy for improved scalability.

**Algorithm 2** Semantic Grouping of Labels (Coarse K-means + Greedy Local Grouping)

---

**Input:** label embeddings $\{(\ell, e_\ell)\}_{\ell=1}^N$, tail subset $\mathcal{S}$, group size $G$, bucket factor $\beta$
**Output:** groups $\mathcal{G}$
**if** $\mathcal{S}$ is provided **then**
  Filter to $\{(y_i, e_i) : y_i \in \mathcal{S}\}$
**end if**
$N \leftarrow$ number of selected labels
$C \leftarrow \max\{1, \lfloor N/(\beta G)\rfloor\}$
Run MiniBatchKMeans with $C$ clusters on $\{e_i\}$ to get cluster IDs $c_i$
Initialize $\mathcal{G} \leftarrow \emptyset$
**for** each cluster $c = 1, \ldots, C$ **do**
  $I_c \leftarrow \{i : c_i = c\}$, mark all $i \in I_c$ as unassigned
  **while** some $i \in I_c$ is unassigned **do**
    Pick a random unassigned seed $s \in I_c$
    Compute similarities $\mathrm{sim}(i) = e_i^\top e_s$ for $i \in I_c$
    Let $T \leftarrow$ top-$k$ unassigned indices by $\mathrm{sim}$, $k = \min(G, \#\text{unassigned})$
    Mark indices in $T$ assigned; add group $\{y_i : i \in T\}$ to $\mathcal{G}$
  **end while**
**end for**
**return** $\mathcal{G}$

---

- **ELMO** (Zhang et al., 2025): A quantization-based training framework for extreme classification that enables low-precision (e.g., BF16/FP8) end-to-end training, substantially reducing memory footprint and accelerating training while preserving predictive performance.

- **RIGL** (Evci et al., 2020): A dynamic sparse training baseline that periodically prunes low-magnitude weights and regrows connections using gradient information.

- **SPARTEX** (Ullah et al., 2024): A fixed fan-in sparse XMC baseline in which each label is connected to a fixed number of input features, providing the closest non-grouped sparse baseline to our method.

- **Block sparsity** (Okanovic et al., 2025): A structured sparse baseline that uses block-level regularity to improve hardware efficiency, but imposes more rigid constraints on the sparsity pattern.

- **VENOM** (Castro et al., 2023): A hardware-aware $V{:}N{:}M$ sparse training baseline that imposes structured row-wise sparsity patterns compatible with efficient sparse execution.

- **SPLAT** (Gupta et al., 2025): A specialized sparse-format baseline based on affine-compressed sparse

*Table 7.* Statistics of XMC Datasets with and without Label Features. This table presents a comparison across various datasets, detailing the total number of training instances ($N$), unique labels ($L$), number of test instances ($N'$), average label count per instance ($\overline{L}$), and average data points per label ($\hat{L}$).

| Dataset | $N$ | $L$ | $N'$ | $\overline{L}$ | $\hat{L}$ |
|---|---|---|---|---|---|
| Amazon-670K | 490,449 | 670,091 | 153,025 | 5.45 | 3.99 |
| AmazonTitles-670K | 485,176 | 670,091 | 150,875 | 5.39 | 5.11 |
| Amazon-3M | 1,717,899 | 2,812,281 | 742,507 | 36.17 | 31.64 |
| LF-Paper2Keywords-8.6M | 2,020,621 | 8,623,847 | 2,020,621 | 9.03 | 2.12 |

*Table 8.* Runtime breakdown of average training step time. Output-layer computation is a major bottleneck for standard fixed fan-in sparsity, while our group-shared design substantially reduces its share.

| Dataset | Model | Average step time | Output layer total | Optimizer step | Output layer share |
|---|---|---|---|---|---|
| AmazonTitles-670K | SPARTEX | 284.0 ms | 241.4 ms | 6.1 ms | 85.0% |
| AmazonTitles-670K | HASTE | 63.7 ms | 13.4 ms | 6.2 ms | 21.0% |
| LF-Paper2Keywords-8.6M | HASTE | 148.1 ms | 47.7 ms | 7.6 ms | 32.2% |

rows, originally designed to exploit structured sparsity patterns such as those arising in sparse attention.

To evaluate model performance in extreme multi-label classification, we report metrics that capture both overall accuracy and performance on rare labels. Our primary metric is Precision@$k$ (P@$k$), which measures the correctness of the top-$k$ predictions. To explicitly account for the highly long-tailed label distributions typical of XMC, we additionally report Propensity-Scored Precision@$k$ (PSP@$k$), which reweights contributions from individual labels to emphasize tail-label performance. While recent work has studied complementary aspects such as calibration of top-$k$ confidence scores (Ullah et al., 2025), our focus is on ranking quality and hardware efficiency.

**Precision at $k$ (P@$k$).**   Precision@$k$ is the standard evaluation metric in XMC tasks such as document tagging and product recommendation. It is defined as

$$P@k(y, \hat{y}) = \frac{1}{k} \sum_{\ell \in \text{top}_k(\hat{y})} y_\ell, \qquad (4)$$

where $y$ denotes the ground-truth label vector, $\hat{y}$ the predicted score vector, and $\text{top}_k(\hat{y})$ the indices of the $k$ highest-scoring labels.

**Propensity-Scored Precision at $k$ (PSP@$k$).**   To address the severe label imbalance present in XMC datasets, PSP@$k$ weights each correctly predicted label by the inverse of its propensity score, thereby amplifying the contribution of rare labels:

$$PSP@k(y, \hat{y}) = \frac{1}{k} \sum_{\ell \in \text{top}_k(\hat{y})} \frac{y_\ell}{p_\ell}, \qquad (5)$$

where $p_\ell$ is the propensity score associated with label $\ell$ (Jain et al., 2016).

**Kernel benchmarking metrics.** For kernel-level evaluation, we report wall-clock execution time measured on the GPU, which directly captures the combined effects of memory access patterns, arithmetic intensity, and hardware utilization. We do not report FLOPs or theoretical throughput, as these can be misleading for sparse kernels whose performance is often dominated by memory bandwidth, data reuse, and kernel launch behavior. All timings are measured using synchronized GPU events and averaged over multiple runs to reduce variance.

## B. Output-Layer Runtime Breakdown

To better understand the end-to-end runtime impact of the output layer, we profile the average training step time and decompose it into output-layer computation and optimizer overhead. Table 8 shows that the output layer is the dominant bottleneck for standard fixed fan-in sparsity: in SPARTEX, it accounts for 85.0% of step time on AmazonTitles-670K. In contrast, our group-shared design reduces the output-layer share to 21.0% on the same dataset. On LF-Paper2Keywords-8.6M, the output layer still accounts for 32.2% of step time, highlighting the importance of hardware-efficient output-layer design at extreme label scales. The semantic label grouping in Algorithm 2 is performed once before training and adds only a small preprocessing overhead. On Amazon-670K, grouping accounts for approximately 1.1% of total training time.

## C. Additional Sparse Baseline Comparisons

We provide additional comparisons against representative sparse-training and sparse-format baselines in Table 9. RIGL is included as a dynamic sparse training baseline, VENOM as a hardware-aware $V{:}N{:}M$ sparse training baseline, and SPLAT as a specialized sparse-format baseline. These results further contextualize the proposed group-shared fixed fan-in design against alternative sparse formulations.

*Table 9.* Comparison with additional sparse baselines. Our method consistently outperforms dynamic sparse training and structured sparse-format alternatives on XMC benchmarks.

| Method | P@1 | P@3 | P@5 |
|---|---|---|---|
| | **AmazonTitles-670K** | | |
| RIGL | 42.0 | 37.3 | 34.4 |
| VENOM | 39.5 | 35.0 | 31.6 |
| SPLAT | 40.5 | 36.2 | 33.0 |
| HASTE | **43.0** | **38.7** | **35.5** |
| | **Amazon-670K** | | |
| RIGL | 45.2 | 38.7 | 36.0 |
| VENOM | 41.2 | 36.1 | 32.0 |
| SPLAT | 41.4 | 36.2 | 32.3 |
| HASTE | **48.1** | **43.1** | **39.2** |
| | **Amazon-3M** | | |
| RIGL | OOM | OOM | OOM |
| VENOM | 47.0 | 43.2 | 40.7 |
| HASTE | **52.5** | **49.5** | **47.4** |

SPLAT's affine-compressed sparse row format is most beneficial when rows have geometrically regular nonzero patterns, such as sparse attention masks with windowed, strided, or block-like structure. XMC output layers are different: they form very tall matrices $W \in \mathbb{R}^{L \times H}$ with $L \gg H$, where rows correspond to labels and columns correspond to encoder features. In this setting, learned supports need not follow affine or local structure. Similarly, VENOM-style $V{:}N{:}M$ constraints improve hardware regularity but impose fixed row-wise sparsity patterns within local feature blocks. By contrast, group-shared fixed fan-in shares supports only across semantically related labels, while each support can still be an arbitrary subset of the full feature dimension. This preserves much of the flexibility of fixed fan-in sparsity while reducing metadata and improving feature reuse.

Relative to the SPLAT-style metadata considered in our experiments, our representation is $1.5\times$ smaller on Amazon-3M ($F = 64 = 2G$) and $3\times$ smaller on LF-Paper2Keywords-8.6M ($F = G = 64$), while also yielding stronger predictive performance in Table 9.

## D. Additional Label Grouping Details

We provide additional details on the scalability and robustness of the semantic label grouping procedure in Algorithm 2. In all main experiments, we use $\beta = 16$ for the intermediate coarse buckets. The group size $G$ is the main structural hyperparameter; as discussed in the main text, it controls the accuracy–efficiency trade-off, while $\beta$ mainly affects grouping runtime rather than final predictive performance. Table 10 confirms this behavior: varying $\beta$ has little effect on Precision@$k$, but substantially changes grouping time.

*Table 10.* Sensitivity to the coarse bucket factor $\beta$ on Amazon-670K. $\beta$ mainly affects grouping time, while predictive performance remains stable.

| $\beta$ | P@1 | P@3 | P@5 | Time (s) |
|---|---|---|---|---|
| 8 | 48.175 | 43.111 | 39.187 | 776.4 |
| 16 | 48.106 | 43.076 | 39.173 | 403.9 |
| 32 | 48.023 | 43.070 | 39.167 | 204.8 |

For the semantic grouping used in the main experiments, we compute instance embeddings with Qwen3-Embedding-0.6B and obtain each label embedding by averaging the embeddings of its positive training instances. To test whether semantic grouping depends on this particular embedding backbone, we repeat the grouping step with two additional pretrained embedding models. The results in Table 11 are very similar across backbones on both Amazon-670K and Amazon-3M, suggesting that the grouping procedure is robust to the embedding model used.

*Table 11.* Effect of embedding backbone used for semantic label grouping. Results are stable across embedding models, suggesting that the grouping procedure is robust to the embedding backbone.

| Embedding model | P@1 | P@3 | P@5 |
|---|---|---|---|
| | **Amazon-670K** | | |
| BAAI/bge-large-en-v1.5 | 48.082 | 43.051 | 39.061 |
| Qwen3-Embedding-0.6B | 48.106 | 43.076 | 39.173 |
| Qwen3-Embedding-4B | 48.107 | 43.081 | 39.171 |
| | **Amazon-3M** | | |
| BAAI/bge-large-en-v1.5 | 52.403 | 49.502 | 47.424 |
| Qwen3-Embedding-0.6B | 52.513 | 49.484 | 47.416 |
| Qwen3-Embedding-4B | 52.597 | 49.543 | 47.452 |

## E. Additional Results on Label-Feature Datasets

We further evaluate our method on additional label-feature XMC datasets in Table 12. These results show that our

*Table 12.* Additional results on label-feature XMC datasets. $M_{tr}$ denotes peak training memory in GiB.

| Method | LF-AmazonTitles-131K | | | | LF-WikiSeeAlso-320K | | | | LF-AmazonTitles-1.3M | | | |
|---|---|---|---|---|---|---|---|---|---|---|---|---|
| | P@1 | P@3 | P@5 | $M_{tr}$ | P@1 | P@3 | P@5 | $M_{tr}$ | P@1 | P@3 | P@5 | $M_{tr}$ |
| SIAMESEXML | 41.4 | 27.9 | 21.2 | 7.1 | 42.2 | 28.1 | 21.4 | 8.9 | 49.0 | 42.7 | 38.5 | - |
| NGAME | 44.7 | 29.9 | 21.2 | 9.0 | 45.7 | 29.6 | 22.1 | 19.3 | 55.0 | 48.1 | 43.1 | 11.03 |
| DEXML | 42.5 | - | 20.6 | 30.2 | 45.1 | 29.9 | 22.3 | 56.1 | **58.4** | - | **45.5** | 75.5 |
| RENEE | **46.1** | **30.8** | **22.0** | 3.0 | **47.9** | **31.9** | **24.1** | 9.1 | 56.0 | **49.9** | 45.3 | 19.9 |
| Dense BN | 39.2 | 25.7 | 18.2 | 2.2 | 44.5 | 28.4 | 21.5 | 3.1 | - | - | - | - |
| RIGL | 43.0 | 28.6 | 20.4 | 3.2 | 44.9 | 29.0 | 21.7 | 9.2 | - | - | - | - |
| SPARTEX | 44.5 | 29.8 | 21.3 | 2.2 | 46.0 | 29.9 | 22.1 | 3.0 | 53.1 | 45.6 | 39.7 | 7.4 |
| HASTE | 44.8 | 30.0 | 21.3 | **1.7** | 47.3 | 31.0 | 23.1 | **2.3** | 55.6 | 49.3 | 45.0 | **5.6** |

method remains competitive with strong XMC, dense, and sparse baselines while using less peak training memory.

## F. Fan-in Ablation

We additionally study the effect of fan-in size $F$ while fixing the group size to $G = 16$. As shown in Table 13, increasing $F$ improves predictive performance, as each label can access a larger subset of input features, but also increases peak training memory and epoch time.

*Table 13.* Effect of fan-in size $F$ with fixed group size $G = 16$. Increasing fan-in improves accuracy at the cost of higher memory and training time. $M_{tr}$ denotes peak training memory in GiB.

| Fan-in | P@1 | P@3 | P@5 | $M_{tr}$ | E. Time |
|---|---|---|---|---|---|
| | | **Amazon-670K** | | | |
| 32 (96%) | 47.2 | 42.0 | 38.1 | 1.9 | 5:32 |
| 64 (92%) | 48.1 | 43.1 | 39.2 | 2.1 | 5:39 |
| 128 (83%) | 49.4 | 44.2 | 40.4 | 2.3 | 5:54 |
| 256 (67%) | 50.2 | 45.0 | 40.8 | 2.6 | 6:15 |
| | | **AmazonTitles-670K** | | | |
| 32 (96%) | 41.2 | 36.8 | 33.1 | 2.9 | 1:31 |
| 64 (92%) | 42.2 | 37.9 | 34.6 | 3.0 | 1:36 |
| 128 (83%) | 43.0 | 38.7 | 35.5 | 3.2 | 1:46 |
| 256 (67%) | 43.6 | 39.2 | 35.9 | 3.4 | 1:58 |

---

**Algorithm 3** Group-Shared Rewiring for Fixed Fan-in Sparsity

---

**Input:** per-label weights $\{w_\ell\}_{\ell\in[L]}$, $w_\ell \in \mathbb{R}^F$; group supports $\{\mathcal{I}_k\}_{k=1}^K$, $\mathcal{I}_k \subseteq [H]$, $|\mathcal{I}_k| = F$; group map $g(\ell) \in [K]$; rewiring fraction $\rho \in (0,1)$; init mode (random/zero)

**Output:** updated supports $\{\mathcal{I}_k\}_{k=1}^K$ and weights $\{w_\ell\}_{\ell\in[L]}$

**Compute group scores.**
Initialize $S \in \mathbb{R}^{K\times F}$
**for** each group $k = 1,\ldots,K$ **do**
  **for** each slot $j = 1,\ldots,F$ **do**
    $S[k,j] \leftarrow \mathrm{mean}_{\ell:\, g(\ell)=k}\,|w_\ell[j]|$
  **end for**
**end for**
**Select pruned slots (global fraction).**
Let $m \leftarrow \lfloor (K\cdot F)\rho \rfloor$
Mark the $m$ smallest entries of $S$ to form mask $M \in \{0,1\}^{K\times F}$
**if** $m = 0$ **then**
  **return**
**end if**
**Regrow group supports.**
**for** each $(k,j)$ such that $M[k,j] = 1$ **do**
  Sample $u \sim \mathrm{Unif}\{0,\ldots,H-1\}$ and set $\mathcal{I}_k[j] \leftarrow u$
**end for**
**Reinitialize weights for affected labels.**
**for** each label $\ell \in [L]$ **do**
  **for** each slot $j = 1,\ldots,F$ **do**
    **if** $M[g(\ell),j] = 1$ **then**
      **if** init mode = random **then**
        $w_\ell[j] \sim \mathrm{Unif}(-1/\sqrt{F},\, 1/\sqrt{F})$
      **else**
        $w_\ell[j] \leftarrow 0$
      **end if**
    **end if**
  **end for**
**end for**

---

