# OpenReview forum: "HASTE: Hardware-Aware Dynamic Sparse Training for Large Output Spaces"
_ICML.cc/2026/Conference — ICML 2026 regular_

### Official Review · Reviewer_N2E1 · 2026-02-25

**Soundness:** 3
**Presentation:** 3
**Significance:** 2
**Originality:** 2
**Overall Recommendation:** 4
**Confidence:** 1

**Summary:**

In extreme multi-label classification with millions of labels, the output layer is a major memory and compute bottleneck. Existing sparsity methods suffer from irregular memory access or poor hardware utilization. The proposed group-shared fixed fan-in sparsity achieves up to 4.4$\times$ faster forward and 25× faster backward passes vs. standard sparsity.

**Compliance With Llm Reviewing Policy:**

Affirmed.

**Key Questions For Authors:**

- What real-world applications or domains typically use extreme multi-label classification? Could the authors provide concrete examples of models (including their names and parameter configurations) used in practice?

- How are semantically related labels identified for grouping? What criteria or methods guide the choice of the number of label groups?

- Will the code be open-sourced?

- According to Table 1, the epoch time of the proposed method is longer than that of Dense BN. Why is this the case, especially if the method is designed to reduce computation and memory overhead?

- In Section 3.3, the paper mentions splitting labels based on frequency. Does the access frequency of labels change during training, and if so, is the label grouping dynamically updated?

**Limitations:**

yes

**Strengths And Weaknesses:**

## Strengths

The experiments are solid and demonstrate clear performance gains. The co-design of algorithm and hardware is particularly innovative: the proposed group-shared fixed fan-in sparsity pattern improves GPU memory access patterns, better utilizes modern compute units like Tensor Cores, reduces memory footprint, and offers higher expressiveness than block sparsity—leading to accuracy that surpasses strong baselines.

## Weaknesses

The paper lacks a detailed performance breakdown. For instance, in extreme multi-label classification tasks, it would be helpful to see how much of the end-to-end training time is spent on the execution of output layer.

---

> ### Author Rebuttal · Authors · 2026-03-31
>
> We thank the reviewer for insightful feedback; our responses follow:
>
> >W1 (performance breakdown / output-layer cost)
>
> We profiled our method and SPARTEX and will include this in revision. Results show the output layer is a bottleneck: for SPARTEX it accounts for 85.0% of step time on AmazonTitles-670K, whereas our hardware-friendly design reduces this to 21.0%. Even on LF-Paper2Keywords-8.6M, it still accounts for 32.2% of step time, underscoring the importance of efficient output-layer design.
>
> | Dataset | Model | Avg. step time | Output layer total | Optimizer step | Output layer share |
> |---|---|---:|---:|---:|---:|
> | AmazonTitles-670K | SPARTEX | 284 ms | 241.4 ms | 6.1 ms | 85.0% |
> | AmazonTitles-670K | Ours | 63.7 ms | 13.4 ms | 6.2 ms | 21.0% |
> | LF-Paper2Keywords-8.6M | Ours | 148.1 ms | 47.7 ms | 7.6 ms | 32.2% |
>
> Label grouping (Algorithm 2) is done once before training and adds minimal overhead. On Amazon-670K, it accounts for 1.1% of training time.
>
>   ---
>
> >Q1 (real-world XMC applications)
>
>
> XMC arises in applications with millions of outputs, including advertising, search/recommendation, and large-scale text classification. Similar large-label settings also occur in scientific domains, e.g., Gene Ontology-based protein function prediction, where DeepGOPlus (Bioinformatics, 2020) predicts over a large ontology of functions, and in drug-target prediction.
>
> A concrete industrial example is NGAME (WSDM 2023), a web-scale Siamese XMC model for retrieval and ads using a shared DistilBERT-base encoder, per-label 1-vs-all classifiers, and MIPS-based retrieval. Relative to dense models (RENEE) and sparse methods (SPARTEX), our work targets the memory-latency trade-off by improving both efficiency and speed.
>
> Related bottlenecks also appear in large-vocabulary language modeling and LLM decoding: Cut Cross-Entropy (ICLR 2025) reduces the cost of full-vocabulary training losses, while FR-Spec (ACL 2025) and EAGLE-3 (NeurIPS 2025) address expensive drafting under large vocabularies.
>
> ---
>
> >Q2 (semantic grouping / number of groups)
>
> We identify related labels using data-driven embeddings. For each label $ℓ$, we compute an offline embedding $e_ℓ$ as the normalized average representation of training instances for which the label is positive. In our implementation, these representations are obtained with Qwen3-Embedding-0.6B on the positive data per label.
>
> We then group labels to maximize within-group cosine similarity. At XMC scale, we use the two-stage procedure in Algorithm 2: (1) KMeans produces coarse clusters, and (2) within each cluster we greedily form groups by selecting a label and grouping it with its nearest neighbors based on cosine similarity. Table 2 compares semantic vs. random grouping and shows its benefit.
>
> To check that grouping is not tied to a single embedding backbone, we also repeated this step with Qwen3-Embedding-4B and BAAI/bge-large-en-v1.5. As shown below, performance is similar across embedding models, suggesting semantic grouping is robust to the embedding model used.
>
> Amazon-670K
>
> | Embedding model | p@1 | p@3 | p@5 |
> |---|---:|---:|---:|
> | BAAI/bge-large-en-v1.5 | 48.082 | 43.051 | 39.061 |
> | Qwen3-Embedding-0.6B | 48.106 | 43.076 | 39.173 |
> | Qwen3-Embedding-4B | 48.107 | 43.081 | 39.171 |
>
> Amazon-3M
>
> | Embedding model | p@1 | p@3 | p@5 |
> |---|---:|---:|---:|
> | BAAI/bge-large-en-v1.5 | 52.403 | 49.502 | 47.424 |
> | Qwen3-Embedding-0.6B | 52.513 | 49.484 | 47.416 |
> | Qwen3-Embedding-4B | 52.597 | 49.543 | 47.452 |
>
> The chosen group size G determines the number of groups L/G, where L is the number of labels. G balances expressiveness and efficiency; we evaluate G∈{16,32,64} in Table 3, where smaller groups improve accuracy while larger groups are more hardware-friendly (Figure 5).
>
>   ---
>
> >Q3 (code release)
>
> Yes.
>
>   ---
>
> >Q4 (Dense BN runtime vs ours)
>
> Dense BN is a FLOPs-matched dense bottleneck baseline, serving as a hardware-efficiency gold standard where FLOPs reduction translates relatively directly into GPU runtime reduction. However, this efficiency comes with reduced expressiveness and weaker predictive performance than fixed fan-in sparsity (SPARTEX) in Table 1. SPARTEX improves performance, but sacrifices runtime because its sparse accesses are less hardware-friendly, so reduced FLOPs do not translate proportionally into runtime gains.
>
> Our goal is to bridge this gap. our method retains the predictive strengths of fixed fan-in sparsity while making computation much more efficient through group sharing. As a result, it achieves similar or better performance than SPARTEX while operating close to Dense BN in runtime (Figure 1.c), providing strong accuracy and efficiency.
>
>   ---
>
> >Q5 (fixed label frequency / dynamic update)
>
> The label frequency (i.e., positive instances per label) in Section 3.3 is computed once over the full training set and fixed for a given dataset. Accordingly, the head–tail split is determined once before training and remains fixed throughout.

---

> > ### Author Rebuttal · Reviewer_N2E1 · 2026-04-05
> >
> > Thank you for the clarifications. I will take the responses into account in my final review cycle.

---

### Official Review · Reviewer_w8Hc · 2026-03-06

**Soundness:** 3
**Presentation:** 3
**Significance:** 3
**Originality:** 1
**Overall Recommendation:** 4
**Confidence:** 4

**Summary:**

The paper proposes a group-shared fixed fain-in sparsity design to train extreme classification models.  This design leads to efficient GPU execution via custom GPU kernels while reducing memory overhead. They also exploit the long-tail structure of XMC datasets by decomposing the output layer into a small dense head over frequent labels and a group-shared sparse head over tail labels. Extensive experiments are done on 4 XMC datasets with labels in the range of 670K to 8.6M. The proposed approach matches or improves Precision@k compared to prior sparse baselines such as Spartex and narrows the performance gap to dense.

**Compliance With Llm Reviewing Policy:**

Affirmed.

**Final Justification:**

The authors have given a detailed response to my questions in the rebuttal. I appreciate it. Based on the responses, I increase my score.

**Key Questions For Authors:**

1. How are the numbers reported for various methods for LF-Paper2Keywords-8.6M dataset in Table 1?
2. Did you try evaluating the proposed approach on other XMC datasets with label features such as LFAT-131K, LFAT-1.3M etc and if yes, how did the proposed approach do?
3. Does the group size and fan-in affect the performance? If yes, do you have any ablation experiments to support your claim?

**Limitations:**

Yes

**Strengths And Weaknesses:**

**Soundness**:
Strengths:
1. The paper is largely technically sound. It clearly identifies a known limitation of training XMC models specifically as the number of labels are in the order of a few millions. The proposed approach aligns well with the GPU constraints such as memory coalescing, reduced irregular memory access etc. The experiments support the main claim of reducing memory and increasing training throughput without significant decrease in P@k.

Weaknesses:
1. The experiments are conducted on a small number of XMC datasets and that too without label features (except LF-Paper2Keywords-8.6M). Even for this dataset, it is not clear where the numbers in Table 1 are reported from. Secondly, comparison with many state-of-the-art XMC methods are missing.

**Presentation**:
Strengths:
1. The paper is well-written and easy to follow.

Weaknesses:
1. The related work section could be deeper to discuss other approaches that involve multi-stage training, label partitioning etc. and why the proposed approach is better than the existing work.
2. It would be good to add more details about the GPU kernel implementation, memory layout etc.

**Significance and Originality**:
Strengths:
1. The paper addresses a highly relevant problem of training XMC models specifically for labels in the range of a few millions. Such models are used in various applications and trained frequently, hence optimizing training will have significant impact in the real world.

Weaknesses:
1. The proposed approach however lacks novelty and focuses primarily on engineering improvements. The idea is incremental over the prior work.

---

> ### Author Rebuttal · Authors · 2026-03-31
>
> We thank the reviewer for the thoughtful feedback. Responses follow:
>
> >Soundness (missing baselines / LF-Paper2Keywords-8.6M)
>
> Please see the clarification on LF-Paper2Keywords-8.6M and the additional label-feature dataset results below. Since Renee is a strong XMC baseline, we did not include a broader set of XMC methods in Table 1. We agree, however, that adding more XMC baselines would improve context, and will expand these comparisons in the revised paper.
>
>   ---
>
> > Presentation (related work depth)
>
> Thank you. In revision, we will expand the discussion of prior multi-stage, label-partitioning, and related XMC methods, and clarify how our approach differs. Many such methods reduce compute by narrowing the prediction space, but do not directly reduce output-layer memory, and are typically not end-to-end. In contrast, our method is an end-to-end, hardware-aware sparse design that reduces both memory and compute.
>
>   ---
> >Presentation (kernel / memory layout)
>
> An anonymized link to the hand-written CUDA kernel is here: https://justpaste.it/lal5o, details will be added in the revised appendix.
>
>   ---
>
> >Originality (novelty vs prior work)
>
> We respectfully disagree and would like to clarify the paper’s main contributions.
>
> 1.  Our sparse format bridges the gap between predictively strong but runtime-inefficient sparsity (e.g., SPARTEX) and hardware-friendly but less expressive structured sparsity (e.g., block sparse). Rather than imposing structure only for efficiency, we derive it from a characteristic property of XMC, semantic relatedness among labels. This yields a sparse representation that is both task-aligned and system-friendly, improving memory locality and feature reuse while preserving predictive quality better than rigid structured formats.
> 2.  We exploit the long-tailed label distribution of XMC to improve sparse-training . The head–tail split provides a small dense pathway for frequent labels, improving gradient flow.
> 3.  We implement this design with truly sparse forward and backward computation with actual memory savings, with a feature-gradient computation that is particularly well matched to Split-K.
>
> Taken together, our contribution is a joint advance in XMC-aware sparsity, XMC-aware sparse training, and efficient sparse execution.
>
> ---
>
> >Q1 (Table 1 LF-Paper2Keywords-8.6M)
>
> The dense results for LF-Paper2Keywords-8.6M in Table 1 are from the ELMO paper that introduced the dataset (lines 267–268, column 2). The other baselines (Dense BN, SPARTEX, block sparse) were run with our model configuration. We will make this explicit in the revised paper.
>
>   ---
> >Q2 (other label-feature XMC datasets)
>
> Yes. We evaluated our method on LF-AmazonTitles-131K, LF-WikiSeeAlso-320K, and LF-AmazonTitles-1.3M; results appear below, showing that our method remains competitive with strong XMC, dense, and sparse baselines while using less training memory.
>
> LF-AmazonTitles-131K [fan-in=128]
> | Method | p@1 | p@3 | p@5 | Mtr |
> |---|---:|---:|---:|---:|
> | SIAMESEXML | 41.4 | 27.9 | 21.2 | 7.1 |
> | NGAME | 44.7 | 29.9 | 21.2 | 9 |
> | DEXML | 42.5 | - | 20.6 | 30.2 |
> | Renee (dense) | 46.1 | 30.8 | 22.0 | 3 |
> | Dense BN | 39.2 | 25.7 | 18.2 | 2.2 |
> | RIGL| 43.0 | 28.6 | 20.4 | 3.2 |
> | SPARTEX| 44.5 | 29.8 | 21.3 | 2.2 |
> | Ours| 44.8 | 30.0 | 21.3 | 1.7 |
>
> LF-WikiSeeAlso-320K [fan-in=256]
> | Method | p@1 | p@3 | p@5 | Mtr |
> |---|---:|---:|---:|---:|
> | SIAMESEXML | 42.2 | 28.1 | 21.4 | 8.9 |
> | NGAME | 45.7 | 29.6 | 22.1 | 19.3 |
> | DEXML | 45.1 | 29.9 | 22.3 | 56.1 |
> | Renee (dense) | 47.9 | 31.9 | 24.1 | 9.1 |
> | Dense BN | 44.5 | 28.4 | 21.5 | 3.1 |
> | RIGL | 44.9 | 29.0 | 21.7 | 9.2 |
> | SPARTEX | 46.0 | 29.9 | 22.1 | 3.0 |
> | Ours | 47.3 | 31.0 | 23.1 | 2.3 |
>
> LF-AmazonTitles-1.3M [fan-in=256]
> | Method | p@1 | p@3 | p@5 | Mtr |
> |---|---:|---:|---:|---:|
> | SIAMESEXML | 49.0 | 42.7 | 38.5 | - |
> | NGAME | 55.0 | 48.1 | 43.1 | 11.03 |
> | DEXML | 58.4 | - | 45.5 | 75.5 |
> | Renee (dense) | 56.0 | 49.9 | 45.3 | 19.9 |
> | SPARTEX | 53.1 | 45.6 | 39.7 | 7.4 |
> | Ours | 55.6 | 49.3 | 45.0 | 5.6 |
>
> ---
>
> >Q3 (group size / fan-in ablations)
>
> Yes. Group size and fan-in affect performance. Table 3 and Figure 5 study G∈{16,32,64}: smaller groups improve accuracy, while larger groups are more hardware-friendly.For fan-in, the ablation below fixes group size = 16 and shows that increasing fan-in improves predictive performance at the cost of more memory and time.
>
> Amazon-670K
> | FANIN (sparsity) | p@1 | p@3 | p@5 | Mtr | E.Ttime |
> |---|---:|---:|---:|---:|:---:|
> | 32 (96%)  | 47.2 | 42.0 | 38.1 | 1.9 | 5:32 |
> | 64 (92%)  | 48.1 | 43.1 | 39.2 | 2.1 | 5:39 |
> | 128 (83%) | 49.4 | 44.2 | 40.4 | 2.3 | 5:54 |
> | 256 (67%) | 50.2 | 45.0 | 40.8 | 2.6 | 6:15 |
>
> AmazonTitles-670K
> | FANIN (sparsity) | p@1 | p@3 | p@5 | Mtr | E.Ttime |
> |---|---:|---:|---:|---:|:---:|
> | 32 (96%)  | 41.2 | 36.8 | 33.1 | 2.9 | 1:31 |
> | 64 (92%)  | 42.2 | 37.9 | 34.6 | 3.0 | 1:36 |
> | 128 (83%) | 43.0 | 38.7 | 35.5 | 3.2 | 1:46 |
> | 256 (67%) | 43.6 | 39.2 | 35.9 | 3.4 | 1:58 |

---

> > ### Author Rebuttal · Reviewer_w8Hc · 2026-04-03
> >
> > Thank you for the detailed answers. Most of my concerns are addressed. However, I am still concerned about the novelty of the proposed approach. I will review any further discussion and consider this limitation with ACs before making my final assessment.

---

> > > ### Author Response · Authors · 2026-04-03
> > >
> > > Thank you for the follow-up and for noting that most concerns were addressed. To clarify the remaining novelty point, we summarize below what we view as the paper’s main contributions.
> > >
> > >
> > > ### **Novelty 1: XMC-aware semi-structured sparsity using label-group support sharing.**
> > >
> > > **Summary:**
> > >
> > > > We propose a semi-structured sparsity formulation for XMC output layers that translates an XMC-aware regularity into a hardware-aware sparse layout that maps naturally to modern accelerator primitives, improving the accuracy–efficiency trade-off relative to prior fixed fan-in and other hardware-efficient sparse baselines.
> > >
> > >
> > > **Details:** A key part of our novelty claim is that, as in several sparse methods, the contribution lies in identifying a sparse pattern that matches the structure of the target workload. Prior work often derives sparse structure from hardware or application regularities, for example row-wise N:M constraints (VENOM[SC'23] ), block structure for efficient tiled execution (BLAST), or workload-specific sparse formats tailored to attention patterns (SPLAT[OOSPLA'25]). In our case, the relevant regularity is that labels can be organized into groups, and using the label group as the sparsity unit enables structured execution in grouped MMA (Tensor Core–compatible) while retaining label-specific weights and much of the flexibility of fixed fan-in.
> > >
> > >
> > > -   **How it differs from SPARTEX:** This does not simply optimize the same formulation. *It changes the sparse representation itself by introducing an XMC-specific prior, which in turn changes both the memory layout and the computation pattern*. This is exactly what enables the gather-once and dense-MMA style execution described in the paper; per-label fixed fan-in does not expose this grouped computation form.
> > >
> > >
> > > -   **How it differs from earlier XMC methods:** While semantic grouping of labels is used in XMC, *the key distinction is its role*. In earlier XMC methods such as Parabel,Bonsai, XR-Transformer semantic/label-space grouping is a divide-and-conquer mechanism, partitioning labels into trees to avoid the linear-in-L cost of one-vs-rest methods such as DiSMEC. By contrast, in our work semantic grouping is used as a hardware mechanism: i.e., to define a shared-support sparse layout that preserves much of fixed fan-in expressiveness while overcoming hardware-efficiency limitations. *Our contribution is therefore not semantic grouping per se, but using semantic grouping to define this sparse layout.*
> > >
> > > -   **How it differs from other sparse formats:** See Q2 (kernel / sparse format comparison) from reviewer iZBy.
> > >
> > >
> > >
> > >
> > >
> > > ### **Novelty 2: Exploiting long-tail label statistics to improve representation learning**
> > >
> > >
> > > **Summary:**
> > > >We exploit the long-tailed label distribution of XMC via a head–tail split: a small dense pathway for frequent labels and a sparse pathway for the tail, providing a consistent gradient signal to the encoder while preserving the memory benefits of sparsity.
> > >
> > >
> > >
> > > - **How it differs from SPARTEX:**  Rather than adding an auxiliary pathway for extra gradient signals early in training, we use label frequency directly to define the dense head + sparse tail. Compared to SPARTEX, this is more memory-efficient, introduces fewer control knobs, and improves performance (Table 6 ablation).
> > >
> > >
> > >
> > > ### **Novelty 3: Truly sparse kernels with Split-K for backward-feature gradients**
> > >
> > >
> > > >This contribution focuses on efficient truly sparse kernel implementation. In particular, we use Split-K for XMC backward-feature gradients; while Split-K (parallelisation over the inner/reduction dimension rather than only outer tiling) is known in high-performance GEMM, its application to XMC backward-feature computation is, to the best of our knowledge, novel in this setting and particularly effective because this step is dominated by a large label-wise reduction, yielding up to ~25× speedup.
> > >
> > >
> > >
> > >
> > >
> > > Given your positive assessment of the paper's technical soundness, hardware alignment, empirical support, and practical relevance, we would be grateful for a bit more clarification on how you are weighing the remaining novelty concern after our response. In particular, it would help us better understand which specific aspect you still view as too close to prior work, since your previous comments otherwise seem to acknowledge a substantive contribution in the problem-specific design and its practical realization.

---

### Official Review · Reviewer_iZBy · 2026-03-15

**Soundness:** 3
**Presentation:** 2
**Significance:** 3
**Originality:** 2
**Overall Recommendation:** 4
**Confidence:** 4

**Summary:**

The paper introduces a fixed fan-in group based sparsification method for learnable weights. This empirical results show this leads to better accuracy (precision) compared block sparse and exhibits good runtime performance over other fixed fan-in methods.

**Compliance With Llm Reviewing Policy:**

Affirmed.

**Final Justification:**

The comparison with related sparse formats was explained in the rebuttal. Other hyperparameter settings were also included.

**Key Questions For Authors:**

The paper tackles the important problem of finding hardware-friendly sparsification techniques for neural networks without compromising too much on accuracy. The proposed group-based fixed-fan in sparsification is nice and can be used to exploit good code generation schemes. That said, there are a few concerns about the technique, evaluation, and related works that led to my final score. Please clarify during the author response.

First, it is unclear to me how you select the fixed clusters as mentioned near line 215 (column 2). This is an important detail that may lead to accuracy improvements or degradations. Algorithm 2 needs to be explained a bit more here. Also, what is the cluster size? How do you determine these? There seem to be many hyperparameters in the design. However, I did not see a hyperparameter tuning study or a rationale for selecting the evaluated hyperparameters.

The kernel implementation strategy is a bit convoluted. Can you provide a generated or hand-written CUDA kernel? Also, what exact sparse data structure are you using? There are other sparse data structures that are proposed specifically to exploit certain patterns that are present in models, such as sparse transformers (e.g., Affine-compressed sparse row [SPLAT, OOSPLA'25] and FlexAttention [MLSys'26]). How do those compare with the sparse formats used in this work? I understand that XMC is different, but since this work proposes a sparse representation that is superior to COO/CSR, it is important to discuss other sparse formats used in specialized workloads.

The evaluation is also a bit convoluted. I did not see a fixed-fan (non group-based) method in Table 1. Maybe Spartex is such a method. If yes, please explain and make it easy for the reader to appreciate the results. Also, I did not see block sparsity being evaluated for latency in Figure 4. Please explain.

Overall, I like the overall idea, but many concerns exist and hence my score.

**Limitations:**

Yes

**Strengths And Weaknesses:**

Strengths
* The idea of group-based fixed fan-in is neat
* Good empirical accuracy performance

Weaknesses
* Some baselines are missing from certain baselines; non-uniformity of baselines (leads to cherry-picking)
* Comparison with other specialized sparse formats is missing.
* The initial presentation of the paper is somewhat convoluted and abstract; however, the technique is well-explained.

---

> ### Author Rebuttal · Authors · 2026-03-30
>
> We thank the reviewer for the constructive comments, especially the insightful suggestion to compare against SPLAT.
>
> > W1, W2 (missing baselines / sparse formats)
>
> We thank the reviewer for suggesting more baseline comparisons. Beyond those already in the paper (i) XMC and dense baselines, (ii) SPARTEX as the standard fixed fan-in baseline, and (iii) block sparsity as a structured sparsity baseline, we have now added (iv) RIGL as a strong dynamic sparse training (DST) baseline, (v) VENOM [SC'23] as a more generic N:M sparse baseline, and (vi) SPLAT as a specialized sparse-format baseline. We will add these in the revision and clarify each baseline’s role.
>
> AmazonTitles-670K
> | Method | p@1 | p@3 | p@5 |
> |---|---:|---:|---:|
> | RIGL  | 42.0 | 37.3 | 34.4 |
> | VENOM | 39.5 | 35.0 | 31.6 |
> | SPLAT | 40.5 | 36.2 | 33.0 |
> | Ours  | **43.0** | **38.7** | **35.5** |
>
> Amazon-670K
>
> | Method | p@1 | p@3 | p@5 |
> |---|---:|---:|---:|
> | RIGL  | 45.2 | 38.7 | 36.0 |
> | VENOM | 41.2 | 36.1 | 32.0 |
> | SPLAT | 41.4 | 36.2 | 32.3 |
> | Ours  | **48.1** | **43.1** | **39.2** |
>
> Amazon-3M
>
> | Method | p@1 | p@3 | p@5 |
> |---|---:|---:|---:|
> | RIGL  | OOM | OOM | OOM |
> | VENOM | 47.0 | 43.2 | 40.7 |
> | Ours  | **52.5** | **49.5** | **47.4** |
>
>   ---
>   >W3 (initial presentation).
>
> Thank you. We will make the initial presentation more concrete.
>
> ---
>
> >Q1 (grouping details and hyperparameters)
>
> We divide L labels into L/G final groups, each containing G semantically similar labels. Since L is large, Algorithm 2 uses two stages. First, it forms coarse buckets by running KMeans with approximately L/(βG) clusters (we use β=16). Then, within each coarse bucket (the for / while part), labels are greedily grouped by cosine similarity: we pick an unassigned label, select its top-(G−1)most similar unassigned neighbors, and form one final group. This repeats until all labels in that bucket are assigned.
>
> Thus, G controls final group size and is the main structural hyperparameter, balancing expressiveness and efficiency. We evaluate G∈{16,32,64} in Table 3; smaller groups improve accuracy, while larger groups are more hardware-friendly (Figure 5). By contrast, β only controls the size of the intermediate coarse buckets and mainly affects grouping efficiency, not final performance. Amazon-670K results:
>
> Amazon-670K
> | β  | p@1   | p@3   | p@5   | Time (s) |
> |---:|------:|------:|------:|:----:|
> | 8  | 48.175 | 43.111 | 39.187 | 776.4 |
> | 16 | 48.106 | 43.076 | 39.173 | 403.9 |
> | 32 | 48.023 | 43.07 | 39.167 | 204.8 |
>
> See also our 3rd response to Reviewer N2E1 for semantic-grouping details.
>
> ---
>
> >Q2 (kernel / sparse format comparison)
>
> Hand-written CUDA kernel: https://justpaste.it/lal5o. We will include additional kernel details in the revised appendix.
>
> Figure 2 in the paper shows the sparse format: a parameter tensor (L, FANIN) and a location-index array (L/G, FANIN).
>
> SPLAT’s ACSR format is most beneficial when each row has a moderately long, geometrically regular set of nonzeros, as in sparse attention masks (windowed/strided/blocked patterns). XMC classifier is instead a very tall L×H matrix with L ≫ H: rows are labels, columns are feature dimensions, and there is no natural affine structure in the learned feature supports (see SPLAT results above). Under DST, supports are rewired to discover better topologies, so constraining each row to an affine-compressible family would  shrink the topology search space and reduce representational flexibility. The same concern applies to row-wise structured constraints such as VENOM, which impose fixed per-row patterns that may improve hardware efficiency but can hurt predictive flexibility. By contrast, fixed fan-in is more natural for XMC, since many tail labels do not require the full encoder feature space, and group-shared fixed fan-in further exploits the fact that semantically related labels often use similar feature subsets.
>
> Compressed metadata is one of SPLAT’s main strengths, but for XMC (L >> H) this advantage is less decisive:
>
> $$
> \frac{\text{metadata(ACSR)}}{\text{metadata(ours)}} \ = \ \frac{3L}{(L/G)\ F} \ = \ \frac{3G}{F}
> $$
>
> where F=FANIN. Therefore, our metadata is  smaller whenever 3G > F; concretely, it is 1.5x smaller on Amazon-3M (F = 64= 2G) and 3x smaller on LF-Paper2Keywords-8.6M (F = G=64), while yielding better predictive performance.
>
>   ---
>
> >Q3 (Table 1 / Figure 4 clarification)
>
>
> Yes, SPARTEX is the non-grouped fixed fan-in baseline in Table 1, and we will make this explicit in the revision.
>
> Figure 4 serves a narrower purpose: it isolates the effect of turning standard fixed fan-in into group-shared fixed fan-in, so it compares only Dense, Dense BN, Fixed fan-in, and Group-shared fixed fan-in. Our goal is to retain fixed-fan-in predictive performance while bringing runtime closer to the FLOPs-matched dense bottleneck. We omit block sparsity because it is a different, more rigid structured regime and is less competitive predictively than fixed-fan-in methods in our setting.

---

> > ### Author Rebuttal · Reviewer_iZBy · 2026-04-04
> >
> > The questions are adequately answered.

---

### Decision · Program_Chairs · 2026-04-30

**Decision:**

Accept (regular)

**Comment:**

The paper received consistently positive scores (WA/WA/WA/WA), with reviewers highlighting solid technical soundness, strong empirical results, and practical relevance for extreme multi-label classification. Reviewers also raised concerns related to missing baselines, hyperparameter choices, grouping strategy, and evaluation scope.

The authors provided a rebuttal, which fully or largely resolved those concerns and one reviewer increased the score to WA.  The AC read the paper, rebuttal and reviews and decided to accept the paper.